# Intelligent Optimization Based on a Virtual Marine Diesel Engine Using GA-ICSO Hybrid Algorithm

**Ximing Chen** [1], **Long Liu** [1], **Jingtao Du** [1], **Dai Liu** [1,*], **Li Huang** [2] and **Xiannan Li** [2]

1   College of Power and Energy Engineering, Harbin Engineering University, Harbin 150000, China; chenximing@hrbeu.edu.cn (X.C.); liulong@hrbeu.edu.cn (L.L.); dujingtao@hrbeu.edu.cn (J.D.)
2   Shanghai Marine Diesel Engine Research Institute, Shanghai 200000, China; huangli@csic711.com (L.H.); lixiannan@csic711.com (X.L.)
*   Correspondence: dailiu@hrbeu.edu.cn

**Abstract:** Considering the trade-off relationship between brake specific fuel consumption (BSFC), combustion noise (CN) and NOx emission, it is a difficult task to optimize them simultaneously in a marine diesel engine. In order to overcome this problem, a novel genetic algorithm and improved chicken swarm optimization (GA-ICSO) hybrid algorithm was proposed, where the enhanced Levy flight and adaptive self-learning factor were introduced in this algorithm. Computational comparisons between GA-ICSO and other effective optimization algorithms were performed using four standard test functions, validating the improvements in both accuracy and stability for GA-ICSO. Furthermore, a predictive engine model based on a phenomenological approach was developed and validated. This model coupled the proposed algorithm for the optimization of a marine diesel engine. In the optimization process, five control parameters were selected as design variables, including injection timing (IT), intake cam phasing (ICP), intake valve closing (IVC), intake temperature and pressure. Results show that, a lower objective value can be obtained by GA-ICSO than other widely used optimization algorithms for all the operating conditions. Besides, by comparing the results between the optimal generations and baselines, it could be found that, under the condition of 50%, 75% and 100%load, CN is reduced by 10.7%, 4.9% and 3.9%, NOx is decreased by 15%, 31% and 33%, and BSFC is suppressed by 10.8%, 13.3% and 9.5%, respectively. Finally, heat release rates, noise spectrums, cylinder pressures and temperatures were all employed to discuss the optimization results of a marine diesel engine under different working conditions.

**Keywords:** marine diesel engine; BSFC; CN; NOx; multi-objective optimization

## 1. Introduction

With the development of economic globalization, the shipping industry plays an essential role in transport, and almost 90% of merchant ships are driven by marine diesel engines [1]. The advantages of high efficiency, high power density and reliability allow the marine diesel engines to occupy a dominant position in propulsion power.

In terms of the stringent Tier III regulations and EEDI (Energy Efficiency Design Index), the fuel economy and NOx emissions of marine diesel engines have to be further improved. Apart from that, the radiated noise problem in marine diesel engines, especially combustion noise (CN), is also becoming more and more serious when it moves to high-speed and heavy-load regions. Due to the trade-off relationships among the BSFC, NOx and CN, it is of great importance to balance the BSFC, NOx and CN simultaneously for marine diesel engines, which is also a difficult task.

Variable Valve Timing (VVT) is an efficient approach to optimize the gas exchange process in the engine, which can improve engine performance and pollution emission. Sabaruddin et al. [2] investigated the optimization of the engine by VVT, and found that VVT achieves fewer emissions, better fuel economy, yet higher torque under any working

condition. Bar-Kohany and Sher [3] reported that the BSFC decreases by 13% and maximum power increases by 6% with the application of VVT for an unthrottled spark ignition engine. Additionally, the inlet and exhaust valves timings were optimized by Menzel et al. [4] for obtaining maximum thermal and volumetric efficiency. Apart from VVT, the injection strategy changes the combustion phase and ignition delay, which has a direct influence on the fuel and air premixed prior to ignition. It is well-known that the position of combustion phase is crucial for BSFC improvement [5], and the intensity of premixed combustion is expected to affect NOx emission and CN [6]. By combining the Miller cycle with a proper injection strategy, NOx and BSFC could be reduced simultaneously in marine engine experiments [7]. However, the large Miller cycle application increased the ignition delay and premixed combustion magnitude, which increased the CN and peak pressure dramatically, and affected the health of the engine. Considering the complex trade-off relationships among NOx, CN and BSFC due to the sensitive influences of IT and VVT, the traditional sensitivity tests are difficult to optimize the large amount of those control variables, especially for the marine engines with high experimental costs. Therefore, statistical and intelligent algorithms are introduced to optimize multi-objective characteristics of marine engines with increasing complexity technologies.

According to the efficient development of computational capacity, statistical methods are always applied to the numerical simulation modeling of diesel engines, which is beneficial for the optimal design of engine control parameters. Chen et al. [8] combined non-linear programming by quadratic Lagrangian (NLPQL) with AVL FIRE to optimize the features of injection and combustion chamber geometry. Taghavifar et al. [9] proposed an optimization method based on Design of Experiment (DoE) with integration of the epsilon-support vector regression (SVR) in AVL FIRE, which was used to reduce both spray droplet diameter and NOx emission at the same time. A multiple linear regression model was employed by Gopal et al. [10] to predict emissions and performance of a diesel engine fueled by a blend of ethanol, biodiesel and diesel. In practice, however, considering the application of on-line optimization, a large sample database (high computing time) has to be avoided. Unfortunately, although the methods mentioned above have become a powerful tool in the optimization, they cannot guarantee their accuracy of results with a small individual size.

Aiming at this issue, in recent years, with the development of bio-inspired algorithms, intelligent optimization methods, such as genetic algorithm (GA), particle swarm optimization (PSO), artificial bee colony (ABC) algorithm, have received considerable attention in the field of engine performance optimization. Shibata et al. [11] optimized the heat release shapes of multiple fuel injections to obtain high ITE and low CN level by means of GA. Wu et al. [12] utilized a micro-genetic algorithm, coupled with an engine computational fluid dynamics code, to optimize a natural gas and diesel dual-fuel engine. However, the basic evolutionary algorithms are easy to fall into the local optima, namely premature phenomenon, which influences the accuracy of the optimization. Hence, improvements on those classical algorithms are introduced by many scholars. Wu et al. [13] designed an adaptive PSO algorithm to identify optimum engine operating points. Hu et al. [14] investigated and optimized seven engine design variables by combining the NLPQL algorithm with multi-objective GA. Zhang et al. [15] presented a hybrid GA-PSO algorithm and applied it to biodiesel engine performance optimization. Cooperative PSO and ABC algorithms are employed by Ogren [16] to find optimal engine operation parameters for triple and quadruple injection schedules. However, even if good benchmark test results are obtained by those newly developed algorithms, the number of iterations is too large, which increases the computational burden, especially for engine optimization. Accordingly, we must propose, with extreme urgency, a powerful optimization method that has the strongest ability to approach the global optima when the iterations number and individual size are both small.

Chicken swarm optimization (CSO) is a new swarm intelligence algorithm developed by Meng et al. [17], which simulates the behaviors of chicken foraging. Compared to a bat

algorithm (BA), GA, PSO, and differential evolution (DE), CSO exhibits better in a lot of popular benchmark test functions [18]. Hence, the effective techniques that help to increase the diversity of swarm could be applied to basic CSO, which will largely overcome the phenomenon of premature convergence and find better solutions than existing algorithms.

In this study, a GA and improved CSO hybrid algorithm (GA-ICSO) was proposed for optimizing BSFC, CN and NOx simultaneously in a marine diesel engine, where the enhanced Levy flight and adaptive self-learning factor was added to make the chicken swarm distributed evenly. Four benchmark test functions were employed to verify the stability and convergence accuracy of the proposed algorithm. Then, the developed GA-ICSO algorithm was combined with a one-dimensional (1D) predictive model for the optimization of a marine diesel engine, and this model was calibrated and validated by phenomenological approach. IT, ICP, IVC and intake pressure and temperature were selected as design variables in the optimization process. Finally, the optimal engine control parameters were obtained in the conditions of 50%load, 75%load and 100%load. The optimized results were compared with baselines and other widely used optimization algorithms, which demonstrates the ability of GA-ICSO algorithm to optimize and balance the CN, NOx and BSFC.

## 2. Preparation

### 2.1. Engine Predictive Model and Validation

Numerical analysis of marine diesel engine was conducted using a commercial one-dimensional engine simulation software, GT-POWER. The numerical model of a marine diesel engine was created, as shown in Figure 1, and validated with the experimental data of the target engine—the specifications of which are listed in Table 1. It is a four stroke, direct injection marine medium-speed diesel engine with a single cylinder. Three operating conditions, including 100%, 75% and 50%load at 1050, 1000 and 850 rpm, have been considered as baseline conditions for the target engine (the experimental data was provided by Shanghai Marine Diesel Engine Research Institute).

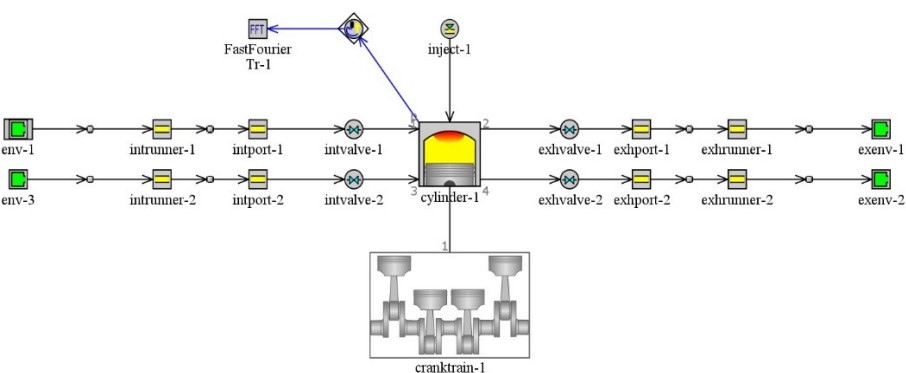

**Figure 1.** Marine diesel engine model.

**Table 1.** Specifications of the engine.

| Item | Specification |
| :---: | :---: |
| Bore | 270 mm |
| Stroke | 330 mm |
| Compression ratio | 16 |
| Connecting rod length | 680 mm |
| Number of strokes | 4 |
| Injection system | Common rail |
| Fuel type | Marine diesel oil |

To predict and analyze the combustion process accurately, the selection of a combustion prediction model in GT-POWER is the most important task. For direct-injection diesel

engines, the direct-injection jet combustion model (DI-Jet) is widely used to predict the combustion rate and emissions [19]. The fuel jet is tracked in DI-Jet model since it breaks into droplets, evaporates, mixes with surrounding gas, and burns by a series of physical or semi-physical equations, as shown in Figure 2. Hence, a DI-Jet model is able to predict the fuel spray and combustion with limited experimental data.

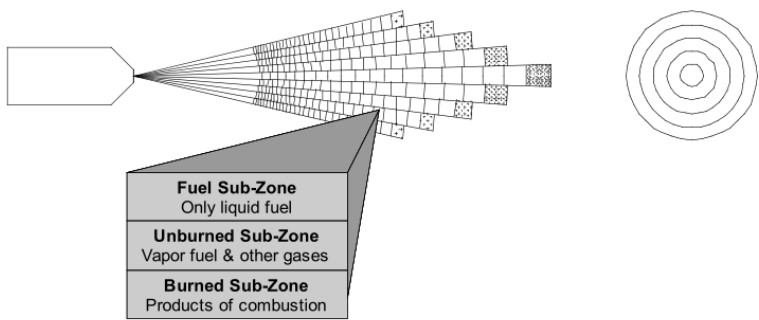

**Figure 2.** Schematic of the fuel zones distribution.

In order to match the combustion event obtained from cylinder pressure analysis, the DI-Jet model has been calibrated based on a phenomenological approach. The experimental data of baseline conditions were employed for calibration in GT-POWER. Figure 3a–c shows the comparisons of experimental (baseline) and GT simulative in-cylinder pressures. It reveals good agreements between experimental data and GT results, which confirms the accurate prediction of the calibrated model.

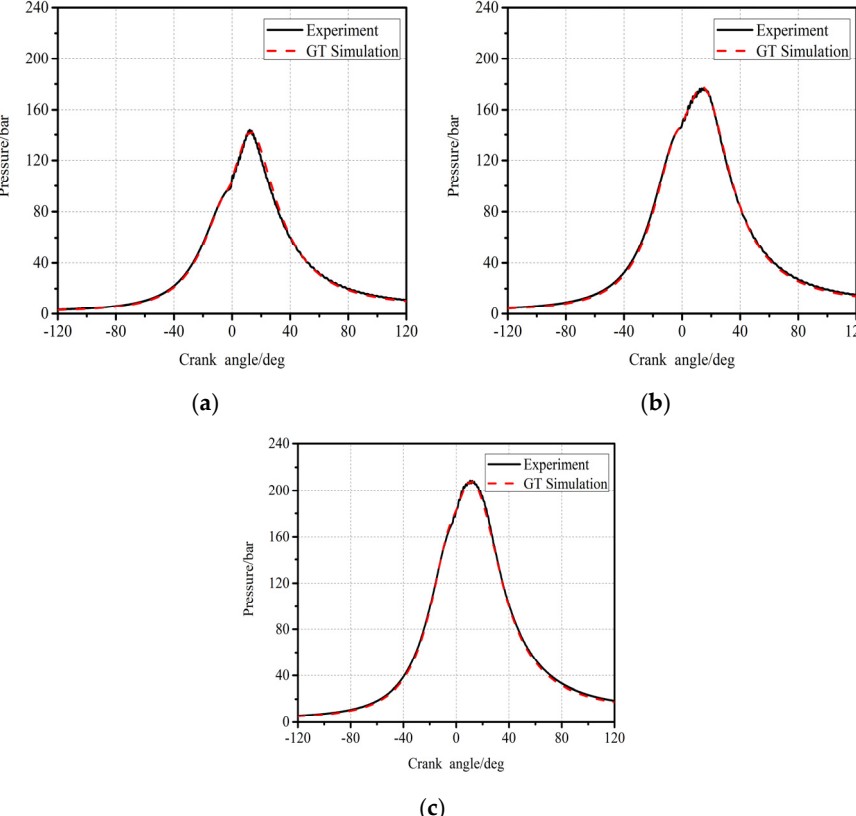

**Figure 3.** Comparison between experimental and GT simulation results of in-cylinder pressures: (**a**) 50%load; (**b**) 75%load; (**c**) 100%load.

Notably, even if the NOx and BSFC can be extracted from the results of GT-Power, the calculations of CN are closely associated with the signal processing technique, which is

unable to be obtained directly from GT-Power. Researchers always utilize maximal pressure rise rate (MPRR) to represent the CN level roughly. However, the recent literature has reported that poor correlation between MPRR and microphone noise is always featured [20] due to the use of different time steps for gradient calculation. Arsham J. [21] proposed a novel method to calculate the combustion noise based on engine structural attenuate curve, Aural weighting (A-weighting) attenuation curve and Fourier function. The engine structure attenuation function can be described as

$$S(dB) = \begin{cases} \sum\limits_{i=0}^{6} a_i f^i & 100 \leq f \leq 2300 \\ \sum\limits_{i=0}^{6} b_i f^i & 2300 \leq f \leq 10000 \end{cases} \tag{1}$$

where $S$ is the attenuation filter function in dB; $f$ is the frequency in [Hz]; $a_i$ and $b_i$ are combustion noise attenuation coefficient, which is listed in Table 2. Notably, the values of $a_i$ and $b_i$ are provided in Ref. [21], and they are defined as filters to simplify the transfer functions of engine structure attenuation. Aside from the structure filtering function, the ear decay filter should also be considered. A-weighting attenuation filter is utilized here due to its excellent ability to model the human ear. The function of A-weighting attenuation can be given by:

$$A(f) = 2 + 20 * \log_{10}(R_A(f))$$
$$R_A(f) = \frac{12200^2 * f^4}{(f^2 + 20.6^2) * \sqrt{(f^2 + 107.7^2) * (f^2 + 737.9^2)} * (f^2 + 12200^2)} \tag{2}$$

where the unit of parameter $A$ is dB. Hence, the total attenuation is established by combining the structure attenuation function and A-weighting attenuation, which is formulated as

$$T(f) = S(f) + A(f) \tag{3}$$

where $T$ is the total attenuation. Figure 4 shows the total attenuate curve. Additionally, Arsham J. discovered that nearly all of engine structural attenuate curves have similar shapes [21], and therefore the attenuate curve in Figure 4 will be used in GT-Power for combustion noise analysis. Figure 5 outlines the procedures of combustion noise calculation in GT-Power. Firstly, the Fast Fourier Transform (FFT) is employed to cylinder pressure signals calculated by GT-Power. Then the combustion noise spectrum can be obtained by applying a total attenuate curve to the pressure spectrum. Finally, the combustion noise spectrum is synthesized as the overall combustion noise level and normalized by 20 μPa, which outputs in dB.

**Table 2.** Combustion noise coefficient.

| $i$ | $a_i$ | $b_i$ |
|---|---|---|
| 0 | $-1.594243 \times 10^2$ | $-1.065899 \times 10^2$ |
| 1 | $2.029175415 \times 10^{-1}$ | $1.89691 \times 10^{-2}$ |
| 2 | $-2.981767797 \times 10^{-4}$ | $-7.393291 \times 10^{-6}$ |
| 3 | $2.494291193 \times 10^{-7}$ | $1.266005 \times 10^{-9}$ |
| 4 | $-1.166026273 \times 10^{-10}$ | $-1.278282 \times 10^{-13}$ |
| 5 | $2.8203001 \times 10^{-14}$ | $7.033316 \times 10^{-18}$ |
| 6 | $-2.747693353 \times 10^{-18}$ | $-1.621458 \times 10^{-22}$ |

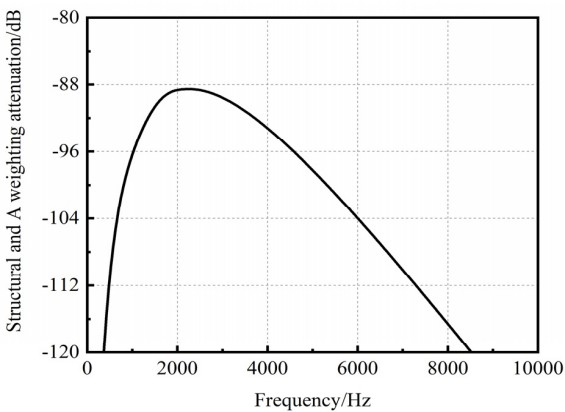

**Figure 4.** Structure and A-weighting attenuation.

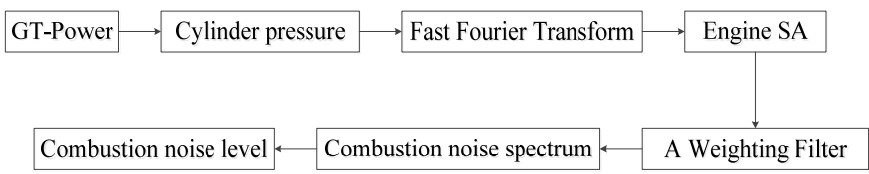

**Figure 5.** Combustion noise level calculation.

As such, the numerical values of BSFC, CN and NOx were compared with that of performance test (baseline conditions), which was revealed in Figure 6a–c and the errors were also shown in Figure 6d. It is found that the simulative data corresponds well with experimental baseline data (within 2% error).

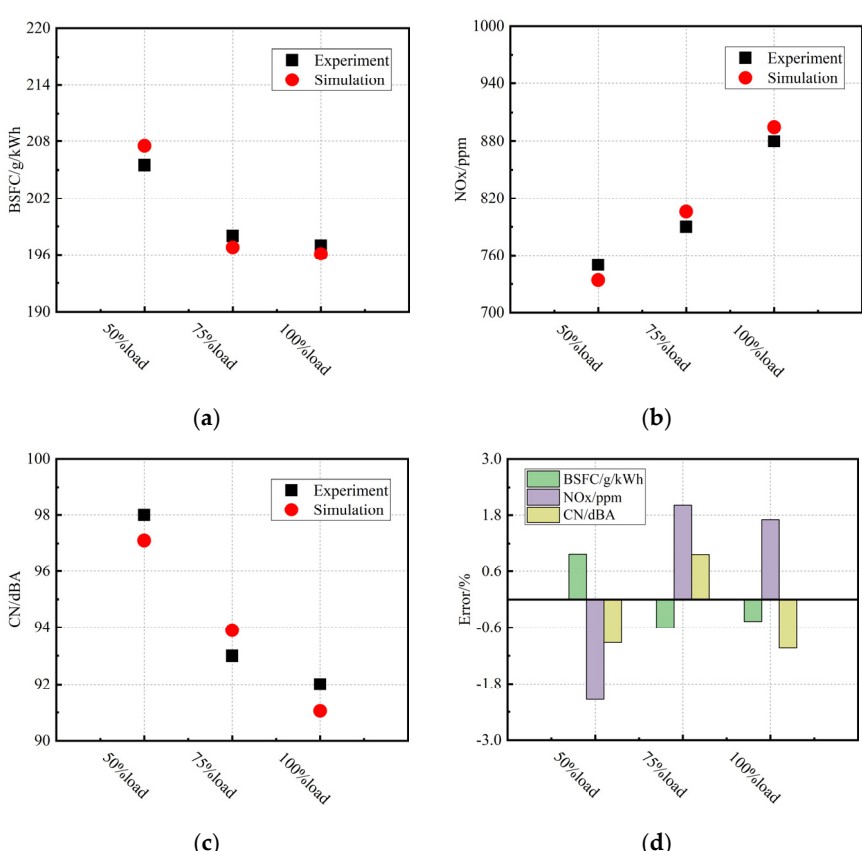

**Figure 6.** Comparisons between experimental and GT simulation results of BSFC, NOx and CN: (**a**) BSFC; (**b**) NOx; (**c**) CN; (**d**) Errors.

*2.2. Objective Function Definition*

With the aim of obtaining the lowest CN, NOx and BSFC simultaneously by the proposed GA-ICSO optimization algorithm, a suitable objective function has to be established, which should consider these three sub-objectives. Equations (1)–(4) define a weighted sum of them, and the optimization of CN, NOx and BSFC could be achieved by lowering the value of *Obj*.

$$Obj = F_{CN} + F_{bsfc} + F_{NOx} + 50 * (\max(1, (\frac{PP}{PP_{limit}})) - 1) \tag{4}$$

$$F_{CN} = a_1 * (\frac{CN - CN_{ideal}}{CN_{base} - CN_{ideal}}) \tag{5}$$

$$F_{BSFC} = a_2 * (\frac{BSFC - BSFC_{ideal}}{BSFC_{base} - BSFC_{ideal}}) \tag{6}$$

$$F_{NOx} = a_3 * (\frac{NOx - NOx_{ideal}}{NOx_{base} - NOx_{ideal}}) \tag{7}$$

where $F_{CN}$, $F_{BSFC}$ and $F_{NOx}$ are the sub-objectives, $[\cdot]_{base}$ and $[\cdot]_{ideal}$ are the baseline and ideal value, respectively. Table 3 shows the ideal values of BSFC, NOx and CN used in this study. The weight of each sub-objective ($a_1$, $a_2$, $a_3$) is determined by researchers according to the literature [8,14] and experience. Table 4 lists the value of them. Considering the high CN level and BSFC for the case of 50%load in Figure 6, $a_1$ and $a_2$ should be given more emphasis and provided with large values in this condition. On the contrary, NOx emission is seriously deteriorated, although a low CN level is observed in the condition of 100%load. So, a large $a_3$, yet a small $a_1$, are suitable for optimization in this case. In addition, the peak value of cylinder pressure (PP) is considered as the constraint variables, and the obtained optimization results, which surpassed the PP constraints ($PP_{limit}$), will be penalized. In this study the $PP_{limit}$ is set as 230 bar.

**Table 3.** Summary of the ideal values in objective function.

| Condition | 50%Load | 75%Load | 100%Load |
|---|---|---|---|
| CN/dBA | 88 | 83 | 81 |
| NOx/ppm | 560 | 690 | 780 |
| BSFC/g/kWh | 195 | 188 | 187 |

**Table 4.** Weight of sub-objective.

| Condition | 50%Load | 75%Load | 100%Load |
|---|---|---|---|
| a1 | 4.5 | 4.4 | 3.8 |
| a2 | 4.5 | 4.4 | 4.7 |
| a3 | 1 | 1.2 | 1.5 |

## 3. Optimization

*3.1. GA*

According to the Darwin theory of evolution, GA is created and used to solve some optimization problems [22]. GA consists of four primary steps: initialization; selection; crossover; and mutation. During the optimization process, a group of individuals (parents) is optimized by GA to generate better ones (children) in the next iteration. When the stopping criterion of iteration is met, the best individual in the last generation is treated as the optimal solution. The feasible solution is named as a chromosome, and each input variable inside the chromosome is called a gene. The steps of GA can be briefly introduced below:

1.  The initial population is created by random generation from the search domain. Additionally, the fitness value of individual is calculated by the objective function;
2.  The individuals with high fitness values should be selected and retained in the next generation, and the others will be eliminated, which simulates the phenomenon of "survival of the fittest";
3.  This operator generates a new individual by exchanging and recombining some genes from two parent chromosomes, which are selected randomly from the population. In this study, the single-point crossover technique is employed;
4.  In the mutation operation, some genes in a chromosome are modified to create a new offspring. This step can increase the diversity of individuals and avoid being trapped into a local optimum.

### 3.2. CSO

The chicken swarm is divided into several subgroups. Each subgroup consists of three types, including a dominant rooster, some hens and several chicks. The flowchart of CSO is shown in Figure 7. A strict hierarchy exists in the chicken group: the chickens with the best fitness values are defined as roosters; the individuals with the worst fitness values can be determined as chicks; the rest of the chickens would be hens. Furthermore, the mother chickens of the chicks are selected randomly. In the foraging process, the rooster has the best food source. Therefore, the hens follow the roosters in each subgroup to search for food, and the chicks follow their mother to forage. The different types of chickens follow different rules of position updated for searching food. Moreover, the hierarchical order and mother–child relationship in each subgroup will not change during several iterations. When the termination condition is met, all the chicken types and their relationships will be reassigned to avoid premature convergence. The optimal solution to the optimization problem can be represented by the position of the best chicken individual, which is retained in each iteration.

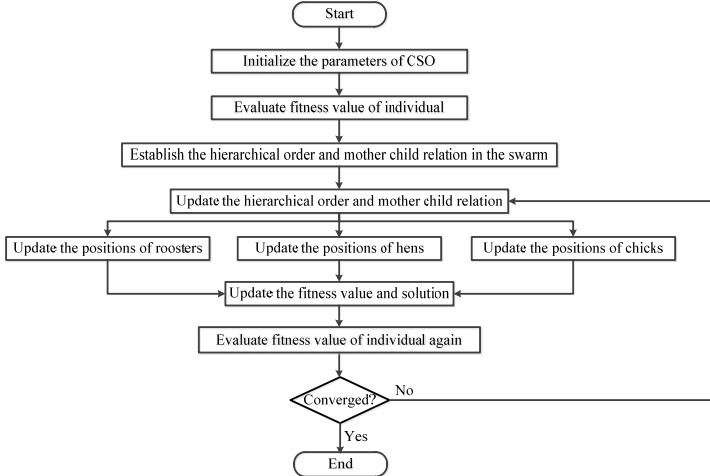

**Figure 7.** Flowchart of CSO.

In this study, *rNum*, *hNum*, *cNum* (=N-rNum-hNum), *mNum* represent the number of the roosters, the hens, the chicks and the mother hens, respectively. *N* indicates the swarm size; *G* is the update number of chicken swarm and *D* is the dimension of solutions space. The maximal iterative generation is *maxiter*. Therefore, the update rules of roosters, hens, and chickens are expressed as follows:

1.  Position update of roosters

$$x_i^j(t+1) = x_i^j(t) * (1 + Randn(0, \sigma^2)), \quad j \in [1, D], t \in [1, maxiter] \tag{8}$$

$$\sigma^2 = \begin{cases} 1, & if \quad f_i \leq f_k, \\ \exp\frac{(f_k - f_i)}{|f_i| + \varepsilon}, & else, \end{cases} \qquad k \in [1, rNum] \ , \ i \in [1, rNum], \ k \neq i \qquad (9)$$

where $x$ is the position of the individual, $i$ and $k$ are both the rooster's index, $j$ is the solution index, $t$ is the iteration number. $Randn(0, \sigma^2)$ is a Gaussian distribution with standard deviation $\sigma$ and mean zero, $\varepsilon$ is a small constant to avoid the denominator being zero. $f_i$ and $f_k$ are the fitness of rooster $i$ and $k$.

2.    Position update of hens

$$x_i^j(t+1) = x_i^j(t) + S_1 * rand * (x_{r1}^j(t) - x_{r1}^j(t)) + S_2 * rand * (x_{r2}^j(t) - x_i^j(t)) \qquad (10)$$

$$S_1 = \exp(\frac{f_i - f_{r1}}{|f_i| + \varepsilon}) \qquad (11)$$

$$S_2 = \exp(f_{r2} - f_i) \qquad (12)$$

where $i \in [rNum + 1, hNum]$, $rand$ is a uniform random number between zero and one. $r_1$, $r_2$ are both the rooster index ($r_1 \neq r_2$), and $r_1$ is the spouse of hen $i$.

3.    Position update of chicks

$$x_i^j(t+1) = x_i^j(t) + FL * (x_m^j(t) - x_i^j(t)), \qquad i \in [rNum + hNum + 1, N] \qquad (13)$$

where $m$ is the index of the mother hen of chick $i$. $FL$ is a random parameter between [0, 2].

*3.3. ICSO*

In order to improve the search ability and accelerate the convergence of CSO, enhanced Levy flight and adaptive self-learning weight are introduced in this work.

1.    Enhanced Levy flight

Levy flight is a type of search method conforming to a short-range deep local search and occasional longer distance walks [23]. The update of Levy flight can be expressed as:

$$Levy(\lambda) = \frac{\lambda \cdot \Gamma(\lambda) \cdot \sin(\pi\lambda/2)}{\pi} \cdot \frac{1}{s^{1+\lambda}} \qquad (14)$$

$$s = \frac{u}{|v|^{1/\lambda}} \qquad (15)$$

$$u \sim N(0, \sigma^2), \qquad v \sim N(0, 1) \qquad (16)$$

$$\sigma^2 = [\frac{\Gamma(1+\lambda)}{\lambda \cdot \Gamma((1+\lambda)/2)} \cdot \frac{\sin(\pi\lambda/2)}{2^{(\lambda-1)/2}}]^{1/\lambda} \qquad (17)$$

where $\lambda$ is the Levy scaling parameter ($1 < \lambda < 3$, and normally $\lambda = 1.5$), and $\Gamma(\cdot)$ is the gamma function. In fact, the foraging movements of many animals are similar to the Levy flight due to its isotropic random directions [24]. Figure 8 compares the traditional random walking following uniform distribution and Levy flight within the same 200 steps. As visible, the search area of random walking is much smaller than that of Levy flight.

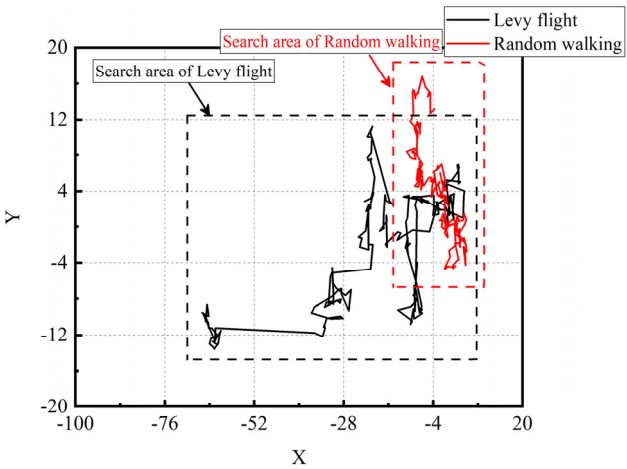

**Figure 8.** Comparison of Levy flight and random walking.

Considering the important role of hens in CSO (largest number), applying Levy flight to the position update of hens is beneficial for search efficiency improvement. Therefore, Equation (3) can be modified as:

$$x_i^j(t+1) = x_i^j(t) + S_1 * rand * (x_{r1}^j(t) - x_{r1}^j(t))$$
$$+ S_2 * rand * (\alpha * Levy(\lambda)) \otimes (x_{r2}^j(t) - x_i^j(t)) \tag{18}$$

where $\alpha$ is an adjustable parameter of flight step, $\otimes$ represents point multiplication. Notably, the parameter $\alpha$ has a significant influence on the speed of search and convergence in basic Levy flight, however, it is always set as a constant. As a result, enhanced $\alpha$ can be formulated as:

$$\alpha(t) = \alpha_{\max} * \exp(c * t) \tag{19}$$

$$c = \frac{1}{maxiter} * \mathrm{Ln}(\frac{\alpha_{\min}}{\alpha_{\max}}) \tag{20}$$

where $t$ and *maxiter* are the current and maximal iteration, $\alpha \in [\alpha_{\min}, \alpha_{\max}]$ ($\alpha_{\min}$ and $\alpha_{\max}$ should be initialized before optimization). The modification of $\alpha$ indicates that short-range deep local search is used when it approaches the optimal solution, which accelerates the convergence.

2.　　Adaptive self-learning coefficient

In the basic CSO, the hens have no self-learning ability, resulting in weakening the search ability of the algorithm to a certain extent [25]. In this section, the adaptive self-learning coefficient is recommended:

$$\omega = \begin{cases} \omega_{\min} + \frac{(\omega_{\max} - \omega_{\min})(f_t - f_{\min})}{f_{\mean} - f_{\min}}, & f_t \leq f_{\mean} \\ \omega_{\max}, & f_t > f_{\mean} \end{cases} \tag{21}$$

where $\omega_{\min}$ and $\omega_{\max}$ are the minimal and maximal self-learning coefficient; $f_{\min}$ and $f_{\mean}$ are the minimal and average fitness value; $f_t$ is the fitness value of each iteration. Clearly in Equation (14), the self-learning coefficient maintains its maximal value when the fitness value is large, illustrating that great search ability is obtained for the early iterations. Furthermore, $\omega$ is set as a very small value when $f_t$ is close to $f_{\min}$, which performs a refined search around the optimal solution. Accordingly, Equation (11), namely the position update of hens, can be further modified as:

$$x_i^j(t+1) = \omega \cdot x_i^j(t) + S_1 * rand * (x_{r1}^j(t) - x_{r1}^j(t))$$
$$+ S_2 * rand * (\alpha * Levy(\lambda)) \otimes (x_{r2}^j(t) - x_i^j(t)) \tag{22}$$

On the other hand, it is seen in Equation (6) that the chicks only update the location information from their mothers and do not refer the roosters. If their mothers fall into a local optimum, the chicks will fall into a local optimum as well. So the position update of chicks is modified as:

$$x_i^j(t+1) = x_i^j(t) + FL * (x_m^j(t) - x_i^j(t)) + C_0 * (x_r^j(t) - x_i^j(t)), \quad i \in [rNum + hNum + 1, N] \quad (23)$$

where $C_0$ is the factor that the chicks learn from the roosters.

### 3.4. GA-ICSO

For the purpose of reducing the probability of the algorithm to fall into the local optimum, GA and ICSO should be combined to integrate the advantages of these algorithms and cover their disabilities. Due to the mutation operator increasing the diversity of individuals, GA is used in the first part of optimization. Then, the individuals optimized by the GA are given to ICSO. So, it is feasible to achieve a fast search of the global optimum by enhanced Levy flight and adaptive self-learning factor in ICSO. Figure 9 reveals the flowchart of GA-ICSO.

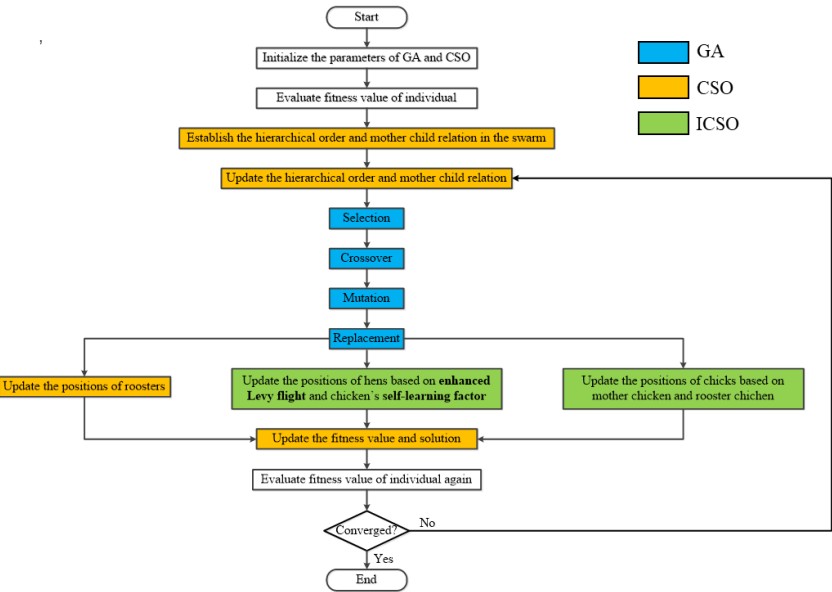

**Figure 9.** The flowchart of GA-ICSO.

In this study, the proposed GA-ICSO is combined with GT-Power for optimizing the engine design parameters, including IT, ICP, IVC, intake pressure and temperature. According to the previous literature [7,11,15], it is demonstrated that these design parameters have a great influence on the combustion process and heat release intensity, which will strongly affect the variations in CN, NOx and BSFC. Their control is commonly applied in current commercial marine engines. The variation ranges in these design parameters and baseline values are all listed in Table 5. At first, the five variables are initialized according to the variation ranges in Table 5, and subsequently used in GT-Power to calculate the objective value by Equation (17). Then the input variables are optimized by the proposed GA-ICSO algorithm and new individuals are generated. The objective values of new individuals are also calculated by GT-Power. Therefore, the optimal individuals (with low objective values) are selected and employed in the next iteration. The flow chart of the hybrid GA-ICSO-GT optimization is shown in Figure 10.

**Table 5.** Variation ranges in the design parameters.

| Parameters | Condition | Lower Bound | Upper Bound | Baseline |
|---|---|---|---|---|
| IT/°ATDC | 50%load | −30 | 10 | −4.5 |
| | 75%load | −30 | 10 | −4 |
| | 100%load | −30 | 10 | −6 |
| ICP/deg | 50%load | −25 | 25 | 0 |
| | 75%load | −25 | 25 | 0 |
| | 100%load | −25 | 25 | 0 |
| IVC/deg | 50%load | 480 | 600 | 540 |
| | 75%load | 480 | 600 | 540 |
| | 100%load | 480 | 600 | 540 |
| Intake pressure/kPa | 50%load | 215 | 225 | 220 |
| | 75%load | 315 | 325 | 320 |
| | 100%load | 384 | 394 | 389 |
| Intake temperature/K | 50%load | 307.5 | 317.5 | 312.5 |
| | 75%load | 308.8 | 318.5 | 313.8 |
| | 100%load | 313.2 | 323.2 | 318.2 |

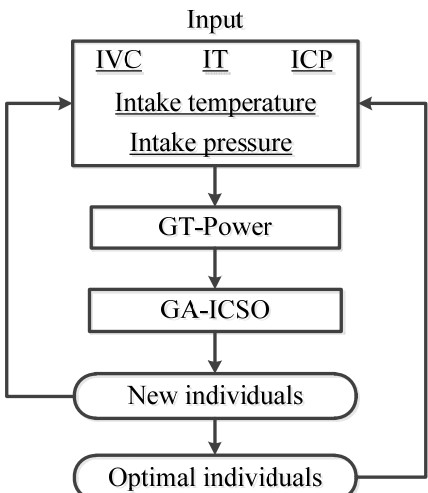

**Figure 10.** The flow chart of the GA-ICSO algorithm combined with GT-Power.

## 4. Benchmark Test

In this section, the performance of the proposed GA-ICSO algorithm is tested on four widely used benchmark functions. The descriptions of these benchmark functions, including function, formulation, minimum and range, are listed in Table 6 [26]. These functions contain a single local optimum (single peak) and many local optima (multi-peak), which are suitable for testing the effectiveness of the optimization algorithms. Furthermore, in Refs. [15,16], it is demonstrated that GA-PSO and an improved artificial bee colony (IABC) show strong abilities in diesel engine performance optimization, although they exhibit poor convergence accuracy and stability when applied to complex multi-input and multi-output problems with a small population size (<20) and low iteration number (<30). So the optimization results of GA-ICSO should be compared to GA-PSO and IABC, which could validate the improvement of optimization accuracy and stability by enhanced Levy flight and adaptive self-learning factor. Meanwhile, in order to validate the improvements of ICSO, GA-CSO and CSO are also run independently for each benchmark function and compared with GA-ICSO.

**Table 6.** Benchmark function details.

| Function | Formulation | Minimum | Range |
|---|---|---|---|
| Schwefel 2.21 | $f(x) = max_{i=1}^{n}(\lvert x_i \rvert)$ | 0 | $[-500, 500]$ |
| Griewank | $f(x) = \frac{1}{4000} \sum\limits_{i=1}^{n} x_i^2 - \prod\limits_{i=1}^{n} \cos(\frac{x_i}{\sqrt{i}}) + 1$ | 0 | $[-100, 100]$ |
| Ackley | $f(x) = -20 \exp(-0.2\sqrt{\sum\limits_{i=1}^{n} \frac{x_i^2}{n}}) - \exp(\sum\limits_{i=1}^{n} \frac{\cos(2\pi x_i)}{n}) + 20 + e$ | 0 | $[-35, 35]$ |
| Zakharovfcn | $f(x) = \sum\limits_{i=1}^{n} x_i^2 + (\sum\limits_{i=1}^{n} 0.5 \cdot i \cdot x_i)^2 + (\sum\limits_{i=1}^{n} 0.5 \cdot i \cdot x_i)^4$ | 0 | $[-2, 2]$ |

The parameter settings of these five optimization algorithms are shown in Table 7. These tuning parameters were selected based on the previous literature [15,18] and empirical evaluations, which were demonstrated to accelerate the convergence of iterations. For example, crossover rate is an important parameter for GA, although an extremely large crossover rate may cause an excessive increase in the randomness, which could disturb the optimization direction towards the global best solution and lose the optimal individual. On the contrary, an extremely small crossover rate may be unable to effectively update the population. Hence, the relatively moderate values of tuning parameters were selected in Table 7. With the aim of validating the global search performance of the proposed GA-ICSO algorithm with a small population, the population size is set to 10 for all the benchmark tests. In addition, the dimensions of all tests are set to be five, which is consistent with the engine design variables defined in Section 3.4. The maximum iterations of each algorithm are set to 50. As such, the average function evaluations and standard deviation (STD) of the results for 100 trials are obtained, as shown in Figures 11 and 12.

**Table 7.** GA-PSO, IABC, CSO, GA-CSO and GA-ICSO control variables.

| Algorithm | Parameters |
|---|---|
| IABC | Limit value = 12, Colony size = 20 |
| GA-PSO | Crossover rate = 0.7, Mutation rate = 0.2, C1 = C2 = 2, $\omega$ = 0.7298 |
| CSO | $G = 30$, $hNum = 7$, $rNum = 2$, $mNum = 2$ |
| GA-CSO | Crossover rate = 0.7, Mutation rate = 0.2, $G = 30$, $hNum = 7$, $rNum = 2$, $mNum = 2$ |
| GA-ICSO | Crossover rate = 0.7, Mutation rate = 0.2, $G = 30$, $hNum = 7$, $rNum = 2$, $mNum = 2$, $C_0 = 0.3$, $\omega_{\min} = 0.2$, $\omega_{\max} = 1.2$, $\alpha_{\min} = 0.5$, $\alpha_{\max} = 1.5$ |

From Figure 11, the proposed GA-ICSO performs higher accuracy than that of other algorithms in finding the objective fitness closest to the global optimization, especially for Griewank and Schwefel function. Moreover, as Figure 12 can be visually seen, compared with other algorithms, the minimal STD value of GA-ICSO demonstrates the stability of the proposed algorithm. Therefore, according to the results of benchmark tests, GA-ICSO is robust in the process of function optimization, which is suitable for the optimization of engine performance and emission.

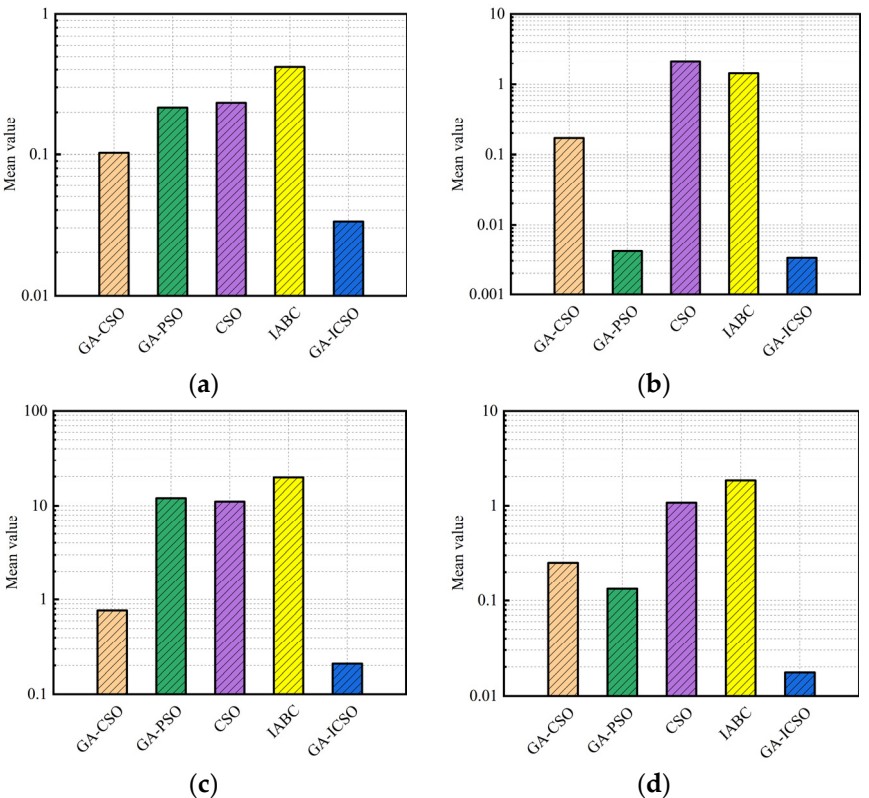

**Figure 11.** Average function evaluations for four test functions: (**a**) Griewank; (**b**) Zakharovfcn; (**c**) Ackley; (**d**) Schwefel 2.21.

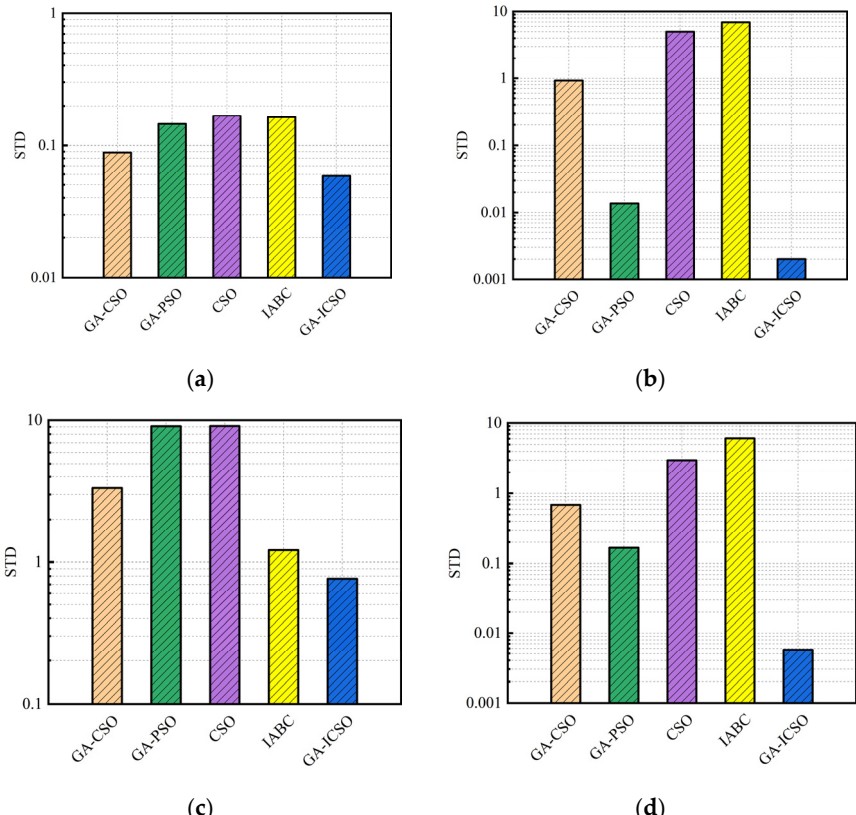

**Figure 12.** STD for 4 test functions: (**a**) Griewank; (**b**) Zakharovfcn; (**c**) Ackley; (**d**) Schwefel 2.21.

## 5. Results and Discussion

Due to the fact that GA-PSO and IABC have been successfully applied to engine optimizations [15,16], both of them are combined with GT-Power for marine diesel engine optimizations in this section, and the results are compared with the proposed GA-ICSO. To reduce the influence of contingency, for each of the three comparison algorithms, the optimization is repeated 10 times independently. The maximum generation is set as 25 and other parameters settings of the algorithms are given to the same values in Table 7. All of the algorithms are applied to three different working conditions: 50%load, 75%load and 100%load.

Figures 13–15 show the optimization history of the best solution and STD of the 10 trails corresponding to three optimization algorithms for the three operating conditions. Observing this progress, the optimizations for all the cases are almost stopped after the 20th generation and the algorithms are considered to have converged. In terms of the baseline values highlighted in each plot, the improvements of overall objective can be observed. Compared with the baselines in Figures 13–15, the objective values optimized by GA-ICSO (25th generation) decrease about 1.04, 0.75 and 0.68 for the conditions of 50%load, 75%load and 100%load, respectively, which is larger than that of GA-PSO (0.74, 0.49 and 0.07) and IABC (0.63, 0.55 and 0.29). It indicates that more reasonable optimization results can be obtained by GA-ICSO. Furthermore, the lowest STD values for GA-ICSO algorithm verifies its stability, and only a small number of independent runs is needed to find the global optimal engine control parameters.

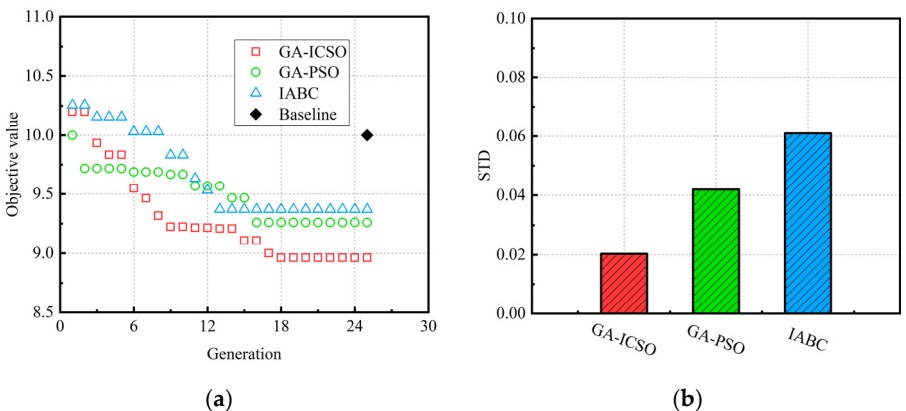

**(a)**                                                          **(b)**

**Figure 13.** Optimization history of GA-ICSO, GA-PSO and IABC for 50%load: (**a**) Objective function convergence; (**b**) STD.

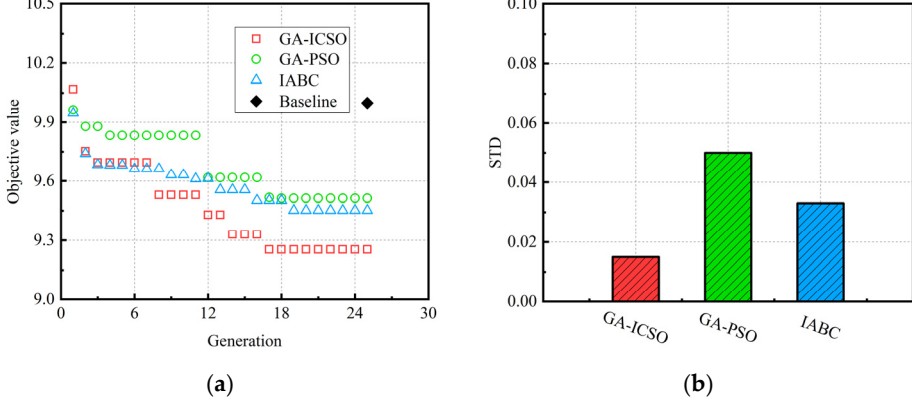

**(a)**                                                          **(b)**

**Figure 14.** Optimization history of GA-ICSO, GA-PSO and IABC for 75%load: (**a**) Objective function convergence; (**b**) STD.

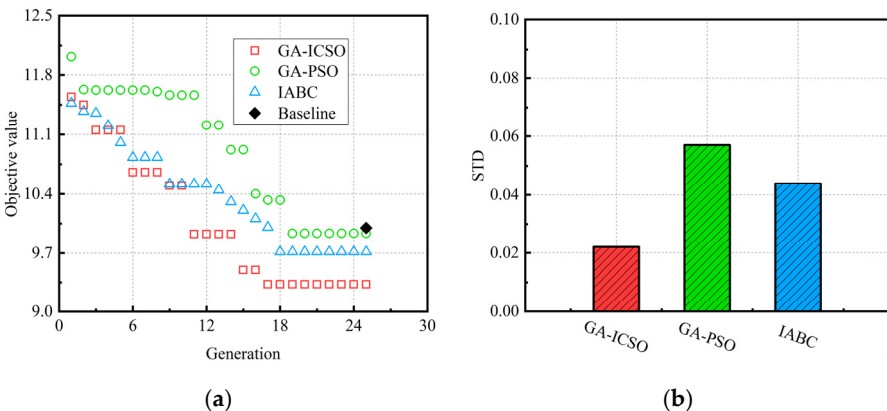

**Figure 15.** Optimization history of GA-ICSO, GA-PSO and IABC for 100%load: (**a**) Objective function convergence; (**b**) STD.

In addition, considering the effects of tuning parameters on optimization results, the values of crossover rate, mutation rate and *G* in GA-ICSO were adjusted for comparative optimization analysis. However, it is found that the obtained optimal solutions are almost unchanged, indicating that the tuning parameters have less of an influence on the optimization results of GA-ICSO. That can be explained by the engine control parameters in baseline conditions having already been roughly optimized by traditional Design of Experiment (DOE) methods, which are used as initial values during GA-ICSO optimization. This can also help to narrow down the optimization range. Therefore, the effects of tuning parameters on optimization process have been weakened.

Subsequently, the inspection of CN, NOx and BSFC is carried out to check the solution success. In order to analyze the optimization results of CN, BSFC and NOx visually, the concept of optimization percentage is defined and expressed as the following:

$$Optimization \quad percentage \quad = \frac{Optimal - Baseline}{|Ideal - Baseline|} \tag{24}$$

where the *Optimal* in Equation (24) illustrates the best CN/BSFC/NOx value optimized by GA-ICSO. Optimization percentage is a negative value, and the lower this value is, the lower the CN/BSFC/NOx level. The optimization percentage of the baseline case is zero. Figures 16a, 17a and 18a compare the optimization percentage of CN, BSFC and NOx for all the conditions, where the red line represents the final solution (25th generation) by GA-ICSO and the black line for the baseline. As visible, CN, BSFC and NOx are improved by GA-ICSO algorithm at the same time. Furthermore, the optimization results of IT, ICP, IVC and intake temperature and pressure are also highlighted in Figures 16b, 17b and 18b. In order to explain the different extent of optimization under different working conditions by thermodynamic analysis, the HRRs, combustion noise spectrums, in-cylinder pressures and temperatures of the optimal generation and baseline for the three working cases are provided, as shown in Figure 16c–f, Figures 17c–f and 18c–f.

For the condition of 50%load, as shown in Figure 16b, IT optimized by GA-ICSO is retarded at about 1.3 CAD (crank angle degree) compared to the baseline, which is the main reason for the premixed combustion attenuation from the HRR plots in Figure 16c. Premixed combustion contributes directly to CN generation, as violent premixed phasing causes rapid pressure rise and induces pressure fluctuations [6]. Accordingly, as seen in Figure 16d, the components of noise spectrum corresponding to the frequency range of 900–1800 Hz are suppressed with the decrease in premixed combustion intensity, and the overall CN level is reduced by about 10.7%.

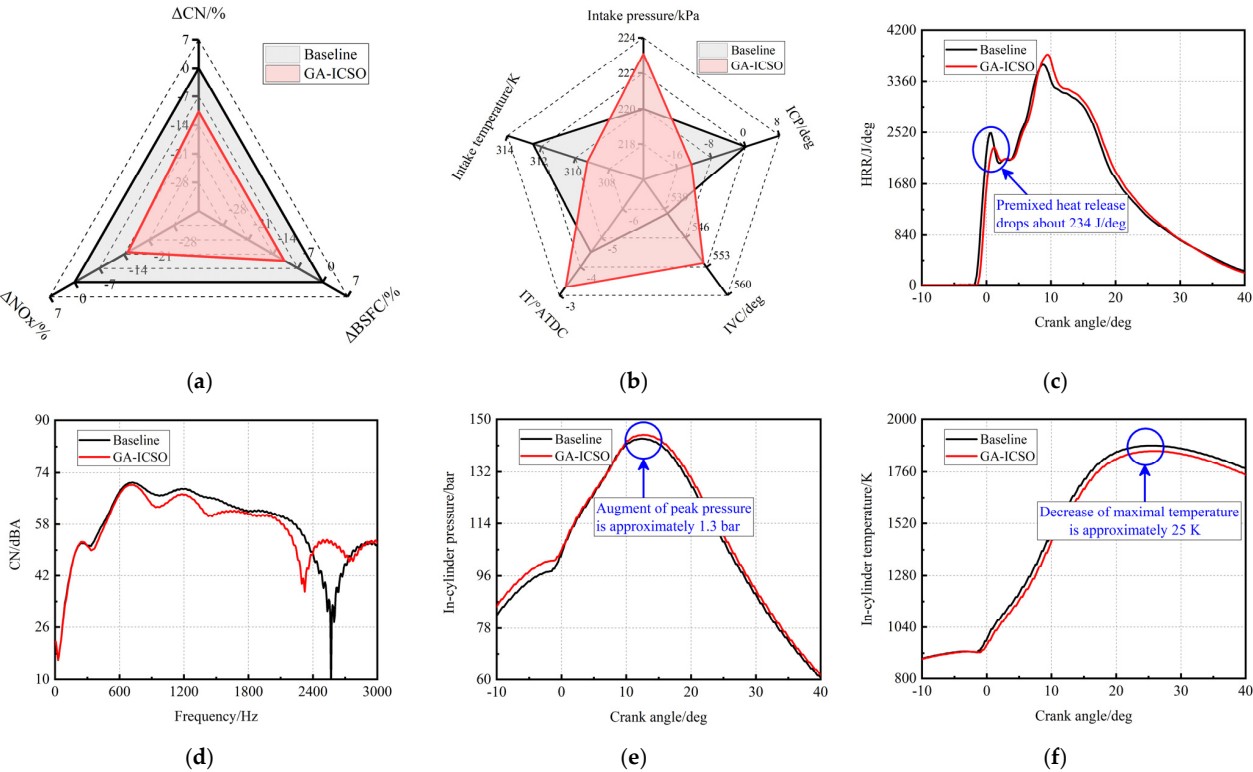

**Figure 16.** Comparisons between optimization results and baseline for the condition of 50%load: (**a**) Optimization percentage convergence; (**b**) Engine control variables; (**c**) HRR; (**d**) Combustion noise spectrum; (**e**) In-cylinder pressure; (**f**) In-cylinder temperature.

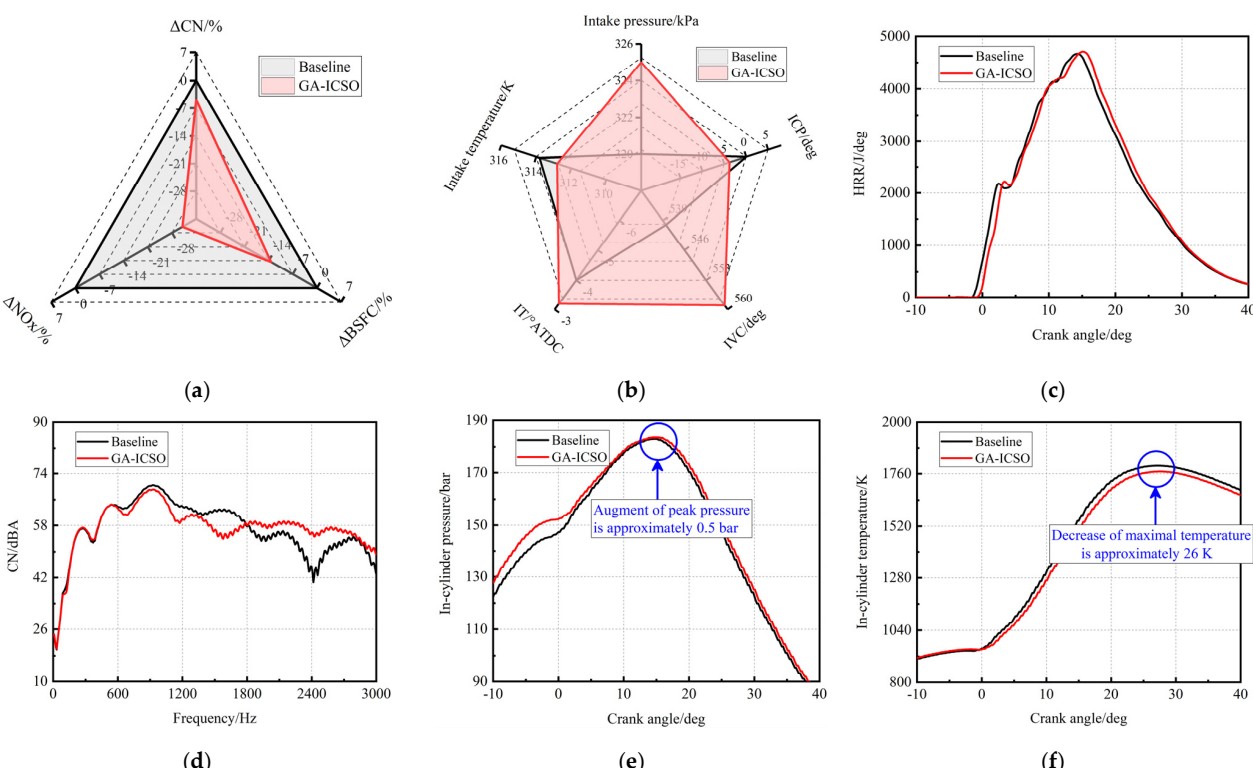

**Figure 17.** Comparisons between optimization results and baseline for the condition of 75%load: (**a**) Optimization percentage convergence; (**b**) Engine control variables; (**c**) HRR; (**d**) Combustion noise spectrum; (**e**) In-cylinder pressure; (**f**) In-cylinder temperature.

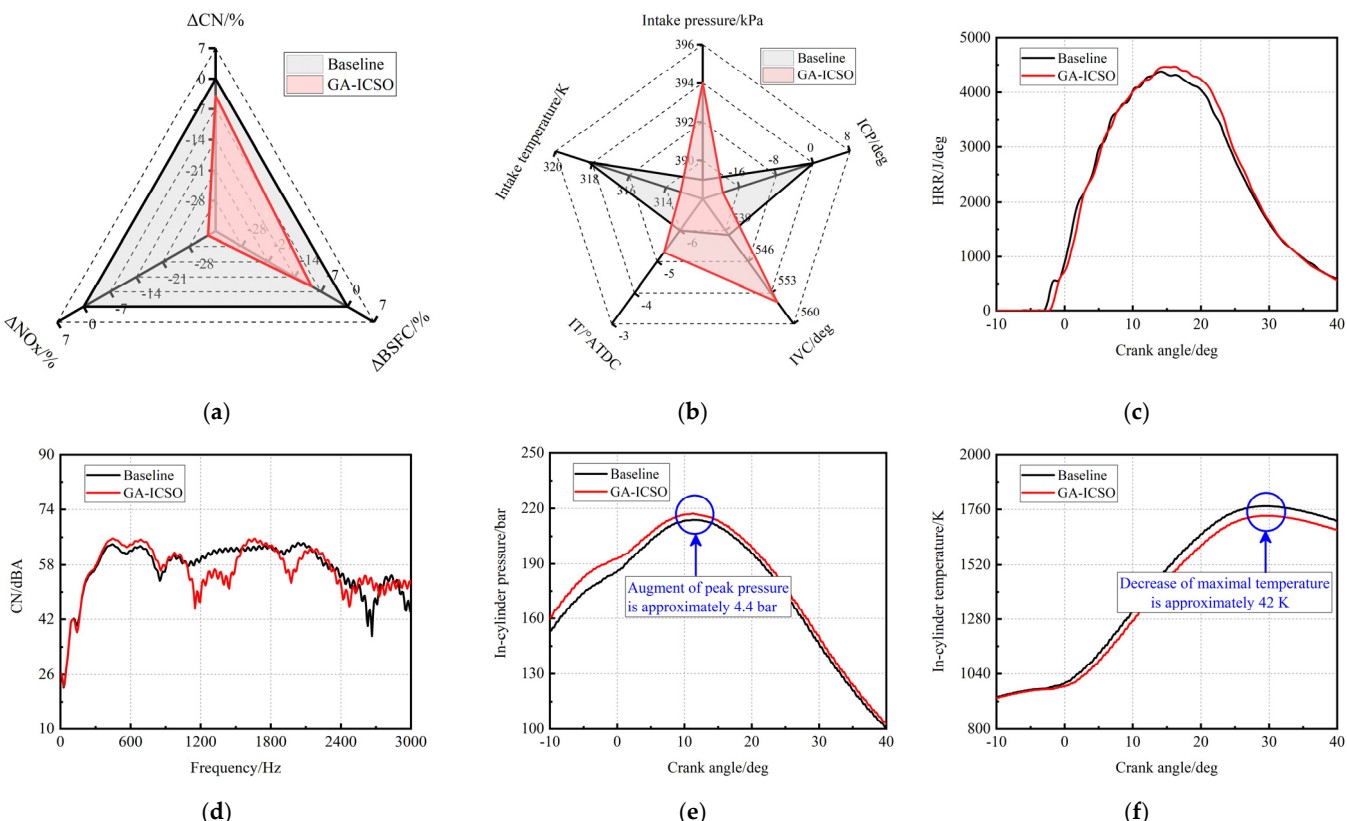

**Figure 18.** Comparisons between optimization results and baseline for the condition of 100%load: (**a**) Optimization percentage convergence; (**b**) Engine control variables; (**c**) HRR; (**d**) Combustion noise spectrum; (**e**) In-cylinder pressure; (**f**) In-cylinder temperature.

However, solely retarding the IT may result in lower values of maximal in-cylinder pressure and temperature, subsequently lower NOx emission yet higher BSFC may be achieved. To overcome the trade-off effect of late IT, the GA-ICSO algorithm proposed an optimized in-cylinder environment: slightly rising the compression-end pressure and maintaining the compression-end temperature, as shown in Figure 16e–f. In this thermodynamic environment, higher air fuel ratio can be achieved and consequently augment the diffusion combustion, as shown in Figure 16c. Thus, even though late IT attenuates the premixed combustion, the BSFC is reduced 10.8 % due to the augment of in-cylinder pressure, as shown in Figure 16e. To achieve the optimized in-cylinder thermodynamic environment, the algorithm suggests adjusting the intake air thermodynamic state and intake valve profile. When compared with baseline condition, the intake pressure and temperature of the optimized case is increased 3.1 kPa and decreased three K, leading to an increase in intake air mass flow rate. On the other hand, advancing ICP (shifts the whole valve profile forward 12.7 CAD, as shown in Figure 16b) enlarges the valve lifts at the start of intake stroke and the valve overlap, which results in larger intake trapped air mass and less in-cylinder residual gas. However, the approaches mentioned above always lead to higher compression pressure. If the compression pressure is over augmented, the peak pressure may exceed the limitation and the temperature may increase as well. Therefore, late IVC is adopted to compensate the effects of other approaches. Finally, the peak value of cylinder pressure rises 1.3 bar in Figure 16e, and the maximal temperature reduces about 25 K in in Figure 16f, which reduce the CN, NOx and BSFC simultaneously.

In engine conditions of 75%load and 100%load, similar optimizations of compression-end pressure and temperature are achieved by the GA-ICSO algorithm. Similar approaches (increasing intake air pressure, reducing intake air temperature, advanced ICP and late IVC, shown in Figures 16b and 17b) are also applied to increase the in-cylinder pressure to an optimized level while maintaining the compression temperature. After optimization,

their increments of compression-end pressure and reductions in maximal temperature are more obvious as the engine load increases, as shown in Figures 16e–f and 17e–f. This is due to the rise of trapped air mass caused by the optimization approaches, leading to higher specific heat capacity. Therefore, even though the optimized IT only delay 0.8 CAD and 0.7 CAD under 75% and 100%load, respectively, their improvements of NOx and BSFC are more significant.

Considering their high loads, it is hard to reduce premixed combustion further due to the relatively short ID. This causes that diffusive combustion phase that occupies the majority of the combustion process, which is unable to generate high frequency pressure fluctuations and exert high CN level [27]. Thus, the optimization percentages of CN in high load regions are always lower than that of low load condition, which is demonstrated in Figures 16a and 18a. Meanwhile, in Figures 17d and 18d, even if the CN levels in the frequency band of 800–1800 Hz are reduced, it is augmented in other frequency segments, e.g., 1800~3000 Hz for 75%load and 400–1000 Hz for 100%load. This gives rise to the inconspicuous optimization of CN, which is only 4.9% and 3.9% for 75% and 100%load, respectively. Finally, the optimized GA-ICSO approach achieved an attenuated or maintained premixed combustion and augmented diffusion combustion (shown in Figures 17c and 18c) in high loads. As such, slight reduction in CN can be achieved with the optimizations of NOx and BSFC.

## 6. Conclusions

In order to optimize BSFC, CN and NOx simultaneously in a marine diesel engine, a novel GA-ICSO algorithm based on enhanced Levy flight and adaptive self-learning factor was proposed. The proposed algorithm was conducted based on 1D predictive model for marine diesel engine optimization. In the optimization process, five control parameters were selected as design variables, including ICP, IVC, IT and intake temperature and pressure. Finally, the optimized results were compared with other widely used optimization algorithms and baselines, which demonstrated the performance of the GA-ICSO algorithm in optimizing engine performances and emissions. The main conclusions are list as follows:

1.  From the results of benchmark tests and engine applications, it is demonstrated that the convergence accuracy and stability of GA-ICSO are higher than other optimization algorithms (including CSO, GA-PSO, GA-CSO and IABC), even for small population size and iteration number;
2.  Due to the trade-off of each parameter, late injection combined with proper in-cylinder environment (slightly raising the compression-end pressure and maintaining the compression-end temperature) is proposed by GA-ICSO algorithm to optimize CN, NOx and BSFC simultaneously;
3.  As the operating condition moves to higher engine load conditions, the mitigation of NOx is much larger than that of CN based on GA-ICSO optimization. As relatively short ID makes the premixed combustion hard to be further suppressed, and therefore the optimization of CN is inconspicuous;
4.  In the optimization of 50%, 75% and 100%load, CN is reduced by 10.7%, 4.9% and 3.9%, NOx is decreased by 15%, 31% and 33%, and BSFC is suppressed by 10.8%, 13.3% and 9.5%, respectively. This is conducted by applying late injection, late IVC timing, early ICP, high intake pressure and low intake temperature derived by GA-ICSO algorithm.

**Author Contributions:** Data curation, X.C.; Formal analysis, X.C.; Funding acquisition, L.L.; Investigation, L.L.; Methodology, J.D.; Project administration, J.D.; Resources, D.L.; Software, D.L.; Supervision, L.H.; Validation, L.H.; Writing—Original draft, X.L.; Writing—Review and editing, X.L. All authors have read and agreed to the published version of the manuscript.

**Funding:** This research was funded by Research of Controllable Combustion Technology in Marine Diesel Engine from National Key R&D Program of China, Ministry of Science and Technology, grant number "2017YFE0116400".

**Institutional Review Board Statement:** Not applicable.

**Informed Consent Statement:** Not applicable.

**Data Availability Statement:** Not applicable.

**Conflicts of Interest:** The authors declare no conflict of interest.

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
