# Peer review of "Intelligent Optimization Based on a Virtual Marine Diesel Engine Using GA-ICSO Hybrid Algorithm"

_machines, doi:10.3390/machines10040227_

Round 1

Reviewer 1 Report

The authors need to strictly address the comments point by point.

  1.    Abstract needs to rewrite with quantitative results.
    2.    Is there are only the top four parameters, injection timing (IT), intake cam phasing (ICP), intake valve closing (IVC), intake temperature and  pressure that affects the CN, NOx, and BSFC. What about compression ratio, load and so on. Furthermore, how you have selected the top four parameters for optimizing CN, NOx, and BSFC.
    3.    In many cases, PSO, JAYA, Rao, and Bald Eagle Search Algorithms, BRANN-TLBO, BRANN-TLBO, BRANN-GWO, BRANN-Jaya resulted in better optimization results. The reason for selection of r GA-ICSO must be strongly justified.
    4.    Improve all Figures that ensure better readability. 
    5.    It is not clear how the Table 2 data has been extracted, and please let us strongly define the parameters ai and bi.
    6.    Please do not abbreviate by yourself use global abbreviation. Example: SA holds good for simulated annealing not structure attenuation.
    7.    Remove flowchart of Fig. 7, as you can see in most of the literature. Shorten the GA description.
    8.    How the tuning parameters of GA-ICSO, GA-CSO, CSO, GA-PSO are fixed.
    9.    How do you ensure your results are generally acceptable for all marine diesel engines.   

Reviewer 2 Report

The paper deals with the development of an intelligent optimization based on a virtual marine diesel engine using GA-ICSO hybrid algorithm.

The scientific goal of this work is present.

Comments

1) Line 14: Please replace be-tween with between

2) Line 16: Please replace further-more with furthermore

3) Line 17: Please replace vali-dated with validated

4) Line 40: Please replace ex-change with exchange

5) Line 43: Please replace un-der with under

6) Line 45: Please replace un-throttled with unthrottled

7) Please provide more technical data regarding common rail system, like manufacturer’s name, pump type, operation pressure

In my opinion, the paper in the present form is not suitable for publication in the Journal of Machines.

Round 2

Reviewer 1 Report

Authors mentioned algorithm optimized parameters are selected from literture review. From past 10years we and our team are working in optimization and we found that there is no universally acceptable algorithm parameters for all the problem domain. The parameters vary from problem to problem. Ensure the selected algorithm parameters are global ones by simulating once again...
